# Signatures of the Auditory Cortex Reveal Discrepancies Across Speech Recognition Models

**Gasser Elbanna**[*]
Speech and Hearing Bioscience and Technology
Harvard University
Boston, MA 02115
gelbanna@mit.edu

**Ivy Brundege**[*]
Department of Brain and Cognitive Sciences
Virginia Tech
Blacksburg, VA 24061
ivybr@mit.edu

**Josh H. McDermott**
Department of Brain and Cognitive Sciences
Massachusetts Institute of Technology
Cambridge, MA 02139
jhm@mit.edu

## Abstract

Speech recognition is central to human communication, yet the neural computations that support it are not fully understood. Artificial neural networks (ANNs) have shown promise as models of sensory systems, and could provide a way to generate candidate hypotheses for the neural representations and mechanisms underlying speech recognition. However, speech-specific ANNs have not been systematically evaluated for this purpose. Here we assess subword-based, word-based, and self-supervised speech models using in-silico simulations of auditory fMRI experiments that probe domain-specific response signatures in human auditory cortex. We find that models optimized for subword units (e.g., phoneme-level) best recapitulate the characteristic patterns of cortical responses, whereas word-level and self-supervised models show worse alignment. These results show how simulations of neuroimaging experiments can reveal facets of model–brain correspondence, providing a complementary diagnostic for refining both speech models and benchmarks of brain–model alignment.

## 1   Introduction

Understanding how the brain transforms sensory input into behavior is a central aim of neuroscience. In audition, speech arrives at the ear as a time-varying acoustic pressure waveform that conveys information about linguistic content, talker identity, and the surrounding acoustic scene. As in other domains of perception, progress in understanding the peripheral and central mechanisms that enable robust speech perception is likely to be aided by stimulus-computable models that reproduce behavioral and neural signatures of speech processing, enabling precise, testable links between theory and data. Such models could arguably be particularly useful for understanding speech, as neuroscience-based approaches are limited by the coarseness of human neuroscience methods.

Recent advances in artificial neural networks (ANNs) have enabled stimulus-computable, sensory-grounded working models that articulate testable hypotheses about function and organization (1). In audition, such models capture human behavior across tasks—word recognition (2; 3), pitch estimation

---

[*]Both authors contributed equally to the paper.

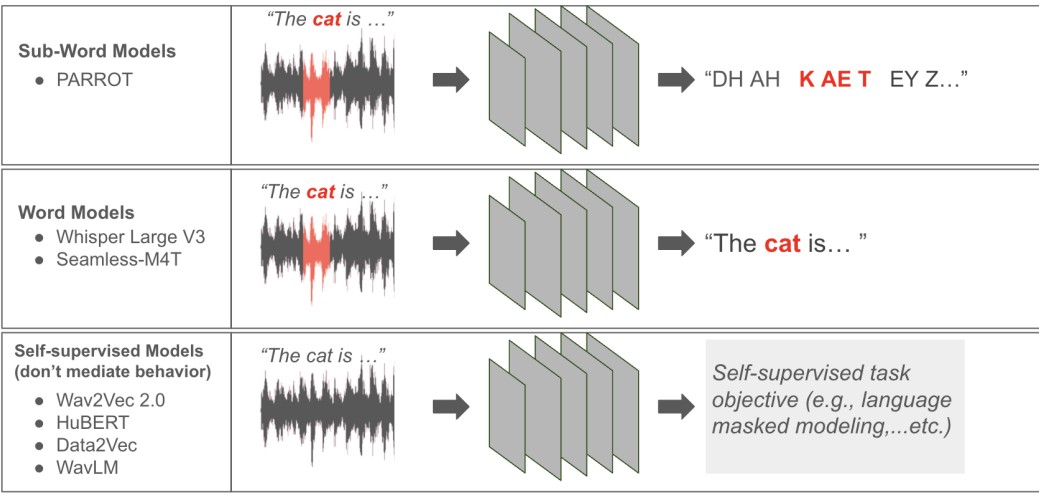

Figure 1: Candidate models spanning three types of speech modeling. Sub-word models are optimized to map acoustic signals to a stream of sub-word units (e.g., phonemes). Word models are optimized to map acoustic signals to a stream of words. Self-supervised models are trained on task objectives that don't require an explicit supervisory signal (e.g., contrastive, language masked modeling, etc.).

(4), sound localization (5), and auditory attention (6)—and explain variance in neural responses throughout the auditory pathway (2; 7; 8; 9).

The most widely used current approaches to assessing human–model alignment fall into two families. Fitting-based methods fit model responses to brain or behavioral measurements (e.g., linear encoding) (10; 11), whereas fitting-free methods compare representational geometry of model and brain responses (e.g., RSA; CKA) (12; 13). While these tools have enabled systematic comparisons, they often reveal similar degrees of human-model similarity across large sets of models (9; 14; 15; 16), motivating the use of additional comparison methods. Here we propose to simulate human fMRI experiments on ANN models, averaging responses within "regions of interest" and comparing responses across conditions as is common in fMRI analysis.

We evaluated three families of speech ANNs—in house subword-optimized models, off-the-shelf word-level models, and off-the-shelf self-supervised models—on their ability to reproduce established fMRI-based signatures of human auditory cortex. We find that these signatures reliably differentiate model classes and reveal clear divergences between some models and brain responses.

## 2 Methods

### 2.1 Candidate speech models

Figure 1 shows the three classes of speech models used in this study. These models are optimized to learn speech representations using different task objectives, architectures, and training diets.

#### 2.1.1 Sub-word models

We trained an in-house sub-word model called PARROT to map acoustic signals to phonemes. PARROT combines a simulation of the human ear with convolutional and recurrent neural network modules. The model was trained on $15,000$ hours of speech data superimposed on naturalistic noisy backgrounds with varying SNRs and sound levels. The experiments described here used representations generated by the last convolution layer and representations from all recurrent layers (6 layers). Model architecture is shown in Supplementary Figure 1.

#### 2.1.2 Word models

We used off-the-shelf state-of-the-art models that are trained to map acoustic signals to words, such as Whisper (17) and SeamlessM4T (18). For each model, we extracted encoder representations from

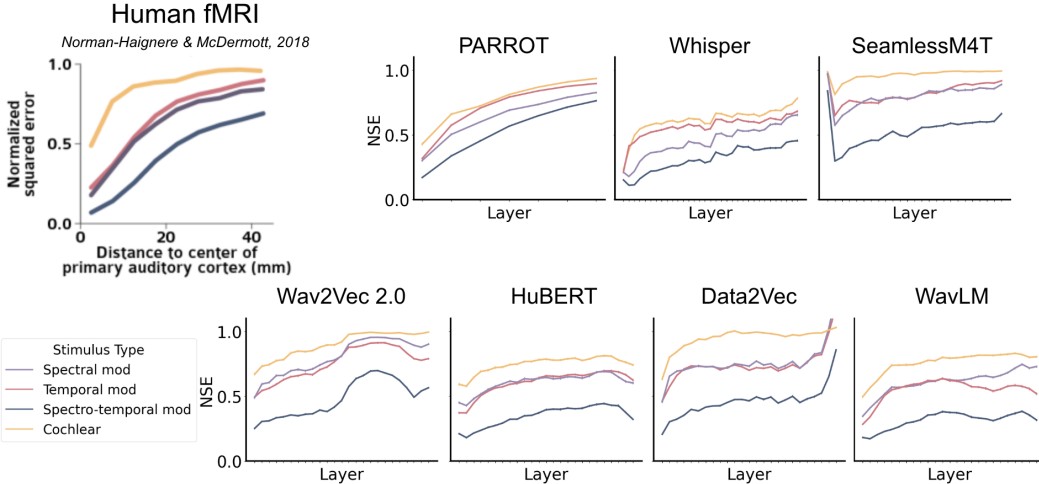

Figure 2: Human and model responses to model-matched stimuli (matched to responses to natural sounds in one of four acoustic models)

the input embedding layer and from every subsequent encoder block. Whisper's encoder comprises 32 Transformer blocks, and SeamlessM4T's speech encoder comprises 24 Conformer blocks.

### 2.1.3 Self-supervised models

We used Wav2Vec 2.0 (19), HuBERT (20), Data2Vec (21), and WavLM (22). We evaluated the large version of these models. The experiments used representations generated by their embedding layer and 24 Transformer encoder layers.

### 2.2 Calculation of model responses

We considered responses of both individual units and entire layers. For a given set of stimuli, a unit response score was defined as the time-averaged absolute values of that unit's response $\text{score}_{\text{unit}} = \frac{1}{T_s} \sum_{t=1}^{T_s} |r_t|$. For a layer's response, the response was further averaged over all units within a layer.

### 2.3 Model-matched stimuli experiment

This experiment replicated the analysis and stimuli from a previous human fMRI study (23).

### 2.3.1 Stimulus generation

For a given natural stimulus, a companion stimulus was synthesized to elicit the same response in a hand-crafted acoustic model intended to capture aspects of biological auditory processing (24). Four acoustic models were used to generate these *"model-matched stimuli"*, varying in the extent and type of acoustic features that were matched across the synthetic and natural sound (cochlear, temporal modulation, spectral modulation, and spectro-temporal modulation). See Supplementary Figure 2 for further illustration.

The cochlear model used a series of bandpass filters designed to mimic the response of the human cochlea (25). The three remaining models pass the filters responses through a series of wavelet filters, convolved with time, frequency, or both, to create models tuned to spectral, temporal, or spectro-temporal modulation, respectively.

### 2.3.2 Response analysis

We calculated the degree of divergence between ANN model responses to original and model-matched stimuli on a unit level, quantified as the normalized squared error (NSE) between natural and its corresponding model-matched response scores:

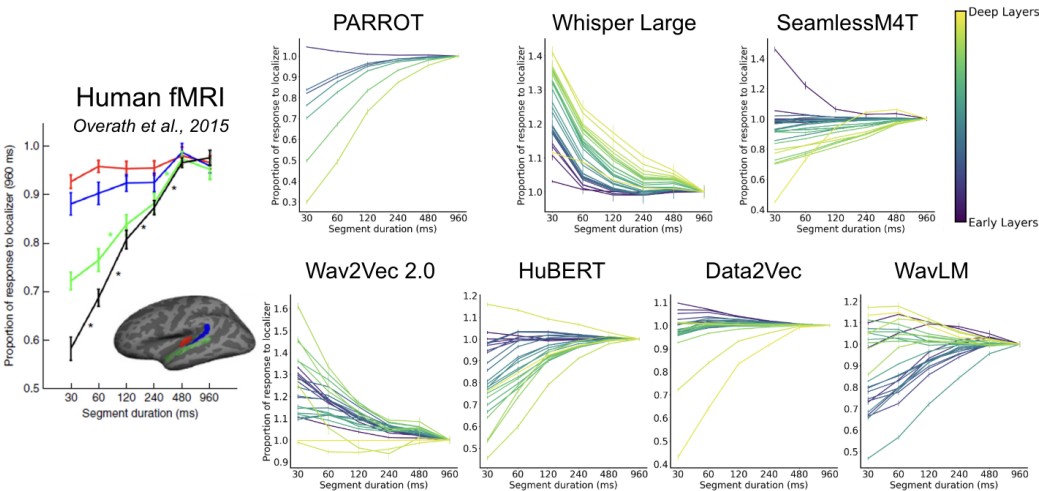

Figure 3: Human and model responses to speech quilts. Speech quilts scramble acoustic structure to different extents depending on the segment length.

$$NSE = \frac{\mu([x - y]^2)}{\mu(x^2) + \mu(y^2) - 2\mu(x)\mu(y)}$$

For each layer, we computed NSE for each unit and report layer-wise responses across models.

## 2.4 Speech quilt experiment

### 2.4.1 Stimulus generation

We used the stimuli first described in (26). Speech quilts are generated by shuffling uniform-length segments of a stimulus, with segments ordered and cross-faded to ensure smooth transitions between segments. Quilts were generated with segment lengths ranging from 30ms-960ms, such that shorter-duration segments lead to greater disruption of the temporal structure of the stimulus. See Supplementary Figure 3 for further illustration.

### 2.4.2 Response analysis

Following (26), we averaged unit responses within a layer, then normalized responses by the average response to the 960ms condition for that layer. We report the average normalized score, by layer, over all segment durations.

## 3 Results

### 3.1 Model-matched stimuli

Figure 2 shows the normalized squared error (NSE) between the response to natural sounds and their corresponding synthesized stimuli produced by four acoustic models, plotted separately for voxels in the human brain and units in candidate models. The NSE is calculated for different brain regions (from primary to non-primary) and different model layers (from early to deep). All models show general trends for a) lower NSE values when spectral and temporal modulation were matched compared to when only a cochlear model was matched, and b) for the NSE to be higher in deeper stages, consistent with the presence of higher-order features that respond more to the natural sounds than to the model-matched sounds. However, the PARROT model most closely resembles what was previously observed in human auditory cortex. See Supplementary Figure 4 for voxel-wise/unit-wise comparison between auditory cortex responses and PARROT model.

### 3.2 Speech quilting

Figure 3 shows brain and models responses to speech quilts in German varying in segment duration (from 30 ms to 960 ms) across brain regions and model layers, respectively. Many of the models are highly discrepant with the responses observed in humans, but PARROT and Data2Vec qualitatively replicate the tendency of non-primary (deeper) auditory stages to respond more to quilts with longer segments lengths (that have more global speech structure).

## 4   Discussion & Conclusion

We compared three classes of speech models—subword-based, word-based, and self-supervised—using simulations of two auditory fMRI experiments previously conducted in humans. Subword models (e.g., PARROT) recapitulated the key response patterns observed in human auditory cortex, whereas word-level and self-supervised models showed less similarity to human brain responses. The results show how fMRI simulation tests that probe for domain-specific response signatures of the human brain provide a complementary additional diagnostic for candidate models.

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

# A    Supplementary Figures

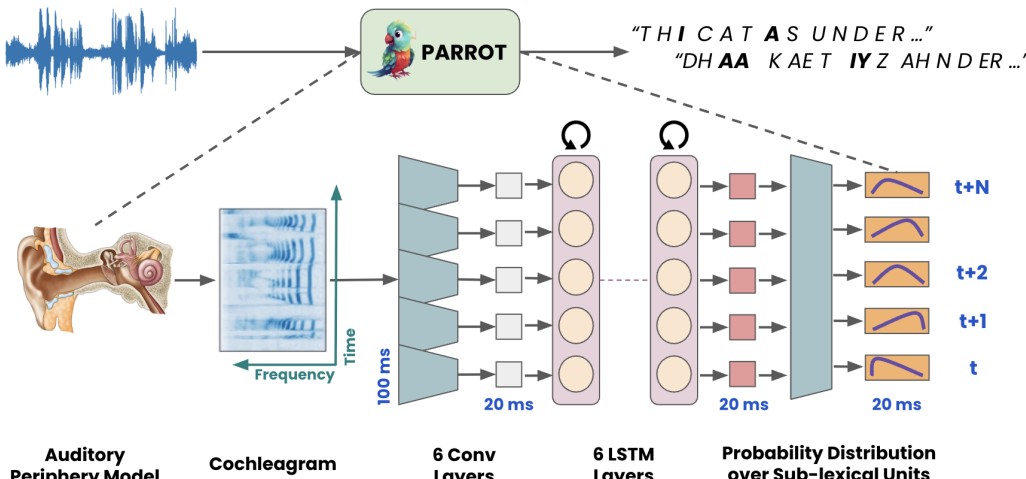

Supplementary Figure 1: Sub-word model (PARROT) architecture.

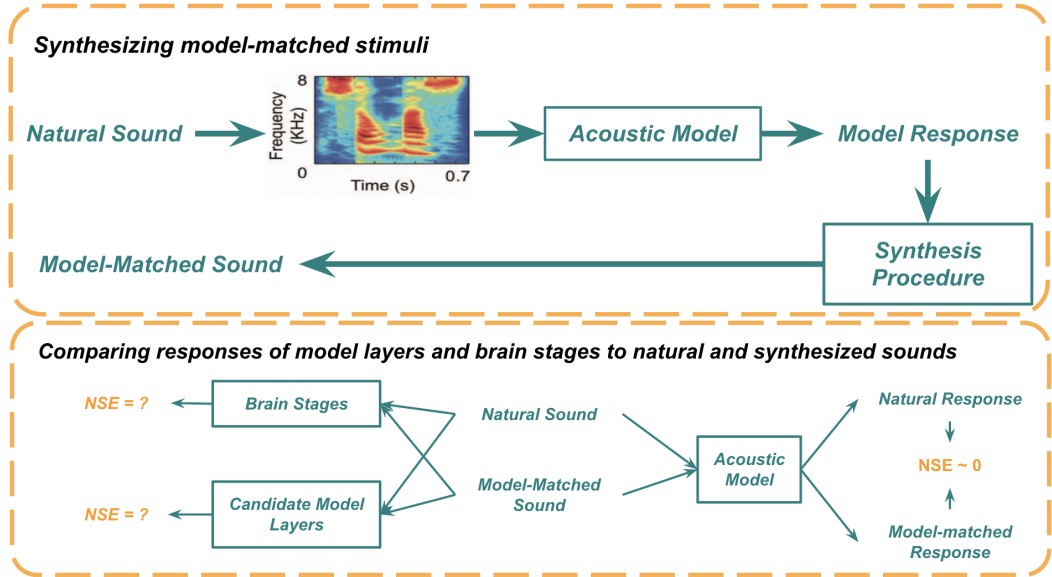

Supplementary Figure 2: Top diagram shows the pipeline for synthesizing sounds that would elicit the same response in a specific acoustic model as the original natural sound. Bottom diagram shows the process for evaluating responses of speech models and brain stages to both natural and their corresponding synthesized sounds. Four acoustic models were used. Thus, for each natural sound, 4 different model-matched sounds were synthesized.

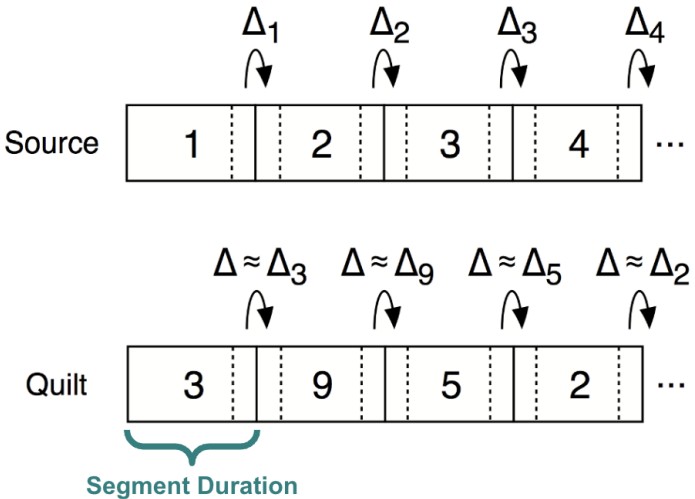

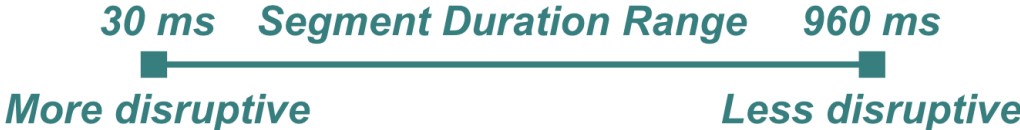

Supplementary Figure 3: Process of generating speech quilts. Figure adapted from (26).

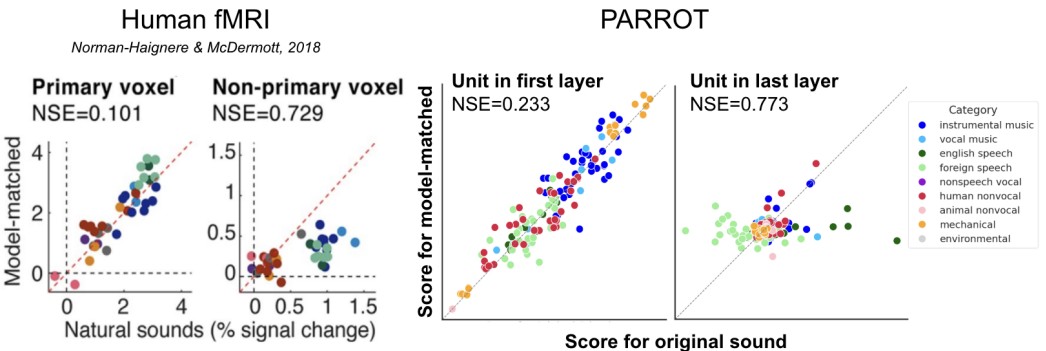

Supplementary Figure 4: voxels and units responses to original and model matched stimuli in the auditory cortex and PARROT model, respectively.

