# OpenReview forum: "Signatures of the Auditory Cortex Reveal Discrepancies Across Speech Recognition Models"
_NeurIPS.cc/2025/Workshop/UniReps — UniReps2025_

### Official Review · Reviewer_7mxo · 2025-09-10
**Good paper but missing important details**

**Confidence:** 3

**Review:**

## Summary
This paper studies alignment between neural activities in neural networks and people when listening to speech, focusing on which kind of network - those that model subwords, words, or unsupervised objectives - best fit the biological data. They find that a subword/phoneme-based network they design fits best, using quilts over a few voxels and model matched stimulus.

---

## Strengths
### Originality
- Most representational alignment work between artificial and biological networks relies on language, not speech models, making the comparison not apples-to-apples, unlike this work. Moreover, the few works that do use speech models ask different questions, e.g. prodding their representation’s semantics [Oota et. al]. Seeing which kind of modelling, especially in the discriminative case, best fits neural representations is interesting as there’s much better task alignment. The authors may want to emphasise this more.
- The PARROT model seems novel, although details on it are lacking.

### Clarity
- I especially like the introduction, where the high level motivation is well laid out.
- The figures are high quality and have a good information density.
- The work itself is well written and very coherent.

### Technical soundness
- A good number of different models are compared and in two different ways, focusing on spatial and temporal similarities (although this isn't explicitly stated, unless I missed it). Although it’s tough making direct comparisons, the authors do their best to do so. Figure 3 is a nice example of this.

### Significance
- There are some subtle points the authors don’t fully emphasize but seem nontrivial, e.g. the comparison between discriminative and unsupervised models, different kinds of discriminative, etc. Discriminative representations should intuitively best align with biological data as only meaningful information is kept, and more useful representations would have a higher magnitude. Thus, this can answer “what task are biological perceptual systems optimized for”, separating semantics from parts of perception (e.g. the quilts vs model matched stimuli showcases this well), parts of speech (as is discussed here), and a myriad of other interesting things.
- The fundamentally different kind of comparison, where it’s functional and not direct representations to representations, is somewhat mentioned in lines 31-34, this should be better emphasized.
---

## Weaknesses
### Originality
I believe this work is original, where I’m unaware and haven’t found other works that try answering these questions via deep learning. There is likely literature I’m unaware of trying to answer this from different angles, e.g. invasive neuroscience studies - they should be discussed. This would help readers coming from deep learning angles better understand the setting, whether these findings are surprising, etc.

### Clarity
- The paper is very high level, with many details omitted both in the main paper and in the brief Appendix. How was PARROT trained? Why isn’t there an existing open source phoneme level model that can be compared to? Explicitly and briefly, what is the paper’s goal? How is the matching done in the model-matching experiment? It’s much harder judging the paper’s quality without these important details.
- Similarly, in many cases there’s a “what” but not a “why” - e.g. in section 2 why are these representations used and not other versions thereof? Why are these design choices chosen? It’s hard judging the paper’s soundness without answers to these questions.
- Many details are omitted. This is likely due to the workshop format but sadly the Appendix doesn’t include them either. For example, to better understand how this work is situated with respect to others I excitedly scrolled down to the Appendix, looking for a related work section, alas it was nowhere to be found.
- Generally, how do the authors think of the paper? I ask as the summary almost makes it seem like a case study which demonstrates a research direction, which I presume it isn’t. What does it show? Does the paper make a prediction that can be tested also irrespective of machine learning models? This is an important point mentioned in the intro but it’s not clear if the paper proposes such a hypothesis.
- Why say “audition” and not “auditory” in the intro? Minor point.
- Would be nice to motivate NSE as a metric, especially for readers coming from classical ML - assuming it’s used to match Norman-Haignere and McDermott, but this should be explicitly stated. Also, it’s currently unclear what mu, x, and y are.
- Figure 2 should have points in addition to the lines as, unlike the human fMRI, layer indices are not continuous.

### Technical soundness
- It’s potentially worrying that the in house model performs best, could it somehow accidentally be biased to the data? For example, overfit to a validation set? Knowing the training setup would help answer this.
- For Figure 2, is layer vs distance the right kind of comparison? The layer likely correlates more with temporal representations than spatial ones, no?
- Similarly, is there a reason we should expect early layers to map onto certain voxels while later ones map onto others? Describing the biological neural pathways would help.
- Why is PARROT the best fit to the data in Figure 2? Is this a qualitative claim? How do we know this isn’t moreso a result of the architecture than of the objective?
- Wouldn’t choosing different ways or layers from which to extract the representations significantly alter the results? These choices should be justified.
- For Figure 3, isn’t it hard making a fair comparison between models due to the different architectures, number of layers, etc.? Ideally the layers should be labelled more than just early->deep.
- Figure 4 in the appendix is missing numbers on its axes.

### Significance
Due to aforementioned clarity issues this is very hard to judge.

### Suggestions/questions/misc

The following paper, while a bit controversial, may have useful constructive criticism for the authors to think of [Schaeffer et. al].

---

## Summary
Although the paper has some very interesting points, sadly it lacks too many details as it is so it’s hard to judge its significance. I implore the authors to resubmit to a different venue after adding the many important details and addressing these comments.

#### References

[Oota et. al] Oota, Subba Reddy, et al. "Speech language models lack important brain-relevant semantics." arXiv preprint arXiv:2311.04664 (2023).

[Schaeffer et. al] Schaeffer, Rylan, et al. "Position: maximizing neural regression scores may not identify good models of the brain." UniReps: 2nd Edition of the Workshop on Unifying Representations in Neural Models. 2024.

**Score:**

2

**Topic Fit:**

3

---

### Official Review · Reviewer_6eEs · 2025-09-14
**This paper evaluates three classes of speech recognition models—subword-based, word-based, and self-supervised—by computationally simulating two auditory fMRI experiments previously conducted in humans. The authors assess these models' ability to reproduce characteristic response patterns of the human auditory cortex. The main findings indicate that subword-based models (e.g., PARROT) best align with human brain responses, while word-based and self-supervised models show less similarity. This work suggests that simulating neuroimaging experiments can serve as a complementary diagnostic tool for refining both speech models and benchmarks of brain-model alignment.**

**Confidence:** 5

**Review:**

Strengths
Novelty in Methodology: The paper introduces a creative and powerful approach to evaluating AI models by using in-silico simulations of fMRI experiments. This methodology is a valuable, complementary metric to traditional performance benchmarks.

Clear and Well-Supported Results: The findings from both simulated experiments clearly demonstrate the strong alignment of the subword-based models with human brain responses. The results are well-structured and easy for the reader to follow.

Potential for High Impact: This research contributes to a better understanding of the relationship between neural network architectures and human brain processes. The findings could inspire future work on designing more brain-aligned AI models.

Weaknesses
Lack of Code and Data: The authors explicitly state in their NeurIPS checklist that the code and data are not available for public use. This is a significant issue, as it not only hinders the reproducibility of the results but also prevents the research community from adopting the in-silico fMRI method as a new benchmark. This is a particularly strong concern for conferences with high standards for reproducibility, such as NeurIPS.

Methodological Limitations:

Limited Generalizability: The study's scope is narrow, examining only a few specific models and simulating just two fMRI experiments. It is unclear whether these findings can be generalized to other models or a wider range of auditory stimuli.

Disconnection from Real-World Performance: The paper focuses on brain-model alignment without explicitly correlating it with the models' real-world performance on standard speech recognition tasks. This leaves a critical gap in understanding whether brain-aligned models are also the most effective ones.

Inherent Limitations of Simulation: While insightful, the in-silico fMRI approach is a simplified approximation of the complex biological processes within the human brain. The simulation may not fully capture all the nuanced neural mechanisms involved in speech processing.

Insufficient Literature Review: The paper lacks a comprehensive review of existing literature. It fails to adequately contextualize its contributions within prior research and does not clearly articulate how its novel approach compares to and improves upon other established evaluation methods.

Recommendation
Based on a thorough review, I recommend that this paper be rated as "Borderline" for acceptance.

Justification: The value of this work is contingent on the workshop's primary objective. If the goal of UniReps is to stimulate discussion and the exchange of preliminary ideas, this paper would be a valuable contribution. However, if the workshop's standard for acceptance is strict reproducibility, as is often the case in top-tier conferences, this paper may not meet the necessary criteria and could be rejected.

**Score:**

3

**Topic Fit:**

3

---

### Official Review · Reviewer_JdtG · 2025-09-15
**Simulated auditory fMRI on model activations shows PARROT best matches cortex, supporting in silico neuroimaging as a diagnostic for brain–model alignment.**

**Confidence:** 3

**Review:**

This paper proposes in silico fMRI diagnostics for brain-to-model alignment in speech by simulating two established auditory cortex paradigms: model-matched stimuli (Norman-Haignere and McDermott, 2018) and speech quilts (Overath et al., 2015), on ANN activations and aggregating units as ROI-like signals. It evaluates a cochlea-inspired, noise-trained subword model, PARROT, word-level ASR models Whisper and SeamlessM4T, and self-supervised encoders Wav2Vec 2.0, HuBERT, Data2Vec, and WavLM. Layer-wise analyses probe canonical auditory signatures. Across both paradigms, PARROT most closely recapitulates human auditory cortex patterns, whereas word-level and most self-supervised models align more weakly; Data2Vec shows a promising trend in the quilt manipulation. Overall, simulated neuroimaging provides a useful diagnostic tool for ANN and brain correspondence.

Strengths
1.    Clear experimental framing grounded in two well-validated fMRI paradigms, yielding interpretable, biologically meaningful readouts for brain–model comparison.
2.    Broad model coverage across objectives and architectures with layer-wise analyses enables informative comparisons across families.
3.    Consistent headline findings: PARROT leads in both paradigms, while Data2Vec shows partial convergence under temporal-structure manipulations in quilts.
4.    Discriminative signals that separate models and layers.
6.    Transparent metrics, using NSE for model-matched stimuli and normalized responses to temporal disruption, aligned with auditory-neuroscience intuitions.

Weaknesses
1.  Confounded central comparison. PARROT differs on three fronts, including a subword objective, a cochlear and modulation-inspired front-end, and noisy training. Because model-matched stimuli live close to that front end in feature space, PARROT has a structural advantage that is agnostic to the objective. The evidence supports this particular subword model rather than the superiority of subword optimization in general.
2.  Simplified neural response modeling and missing pseudo BOLD control. Time-averaged absolute activations pooled within layers discard temporal dynamics and response polarity, so they do not approximate BOLD. Robustness to a pseudo BOLD pipeline is not shown; a proper check would center activations, convolve with a hemodynamic response function, add realistic noise, estimate GLM betas, and rerun the alignment analyses.
3.  ROI to layer mapping is implicit, and coverage is narrow. Figures treat early and deep layers as proxies for primary and nonprimary cortex, yet analyses focus on A1 and Heschl’s gyrus with coarse aggregation elsewhere. A fuller assessment should include bilateral superior and middle temporal gyri and the left hemisphere language network, for example, the inferior frontal gyrus and supramarginal gyrus.
4.  Controls and external validity are limited. Only encoders are analyzed for Whisper and Seamless, although decoder conditioned states may differ. There are no size-matched or front-end matched ablations, and the quilt language in German may not match some models’ training. Prior work often finds strong alignment for self-supervised models, including language ROIs, yet the paper does not reconcile these results or test whether alignment varies by ROI class, such as auditory versus language networks.
5.  Quantitative validation is insufficient. The study reports no statistics tying brain ROI effect profiles to layer effect profiles, such as rank correlations of segment length slopes, and does not control for multiple comparisons. The evidence leans on qualitative pattern matching rather than formal tests or established alignment metrics, for example, noise ceilings and cross-validated encoding performance with effect sizes and confidence intervals.
6.  Methodological details are underspecified. Key elements need clarification, including PARROT’s exact architecture beyond the supplementary figure, the composition of the training data, and the rationale for layer selection in comparison models.

**Score:**

2

**Topic Fit:**

2

---

### Official Review · Reviewer_wrZc · 2025-09-17
**Review 102**

**Confidence:** 4

**Review:**

**Title**: *Signatures of the Auditory Cortex Reveal Discrepancies Across Speech Recognition Models*

**Abstract**: The abstract clearly summarizes the motivation, experimental setup, and main results: subword-based models better capture auditory cortical response signatures than word-based or self-supervised models. It is concise and accurately reflects the scope of the paper, though it could benefit from specifying the number of models tested and explicitly stating the novelty (simulation of fMRI signatures as a diagnostic tool).

---

### Strengths
- **Novel evaluation method**: Simulating fMRI experiments to compare brain–model alignment provides an original and complementary diagnostic beyond conventional encoding/RSA analyses.
- **Comparative breadth**: Includes three model families (subword, word, self-supervised), with well-chosen exemplars (PARROT, Whisper/SeamlessM4T, Wav2Vec/HuBERT/Data2Vec/WavLM).
- **Neurobiological grounding**: Directly leverages established fMRI paradigms (model-matched stimuli, speech quilting), linking computational results to prior neuroscience studies.
- **Clear results**: Demonstrates that subword-optimized models better replicate human auditory cortex signatures, aligning with hypotheses about hierarchical auditory processing.
- **Methodological transparency**: Detailed descriptions of stimulus generation, response analysis, and layer/unit response aggregation.

---

### Weaknesses
- **Limited scope of evaluation**: Only two fMRI paradigms are simulated, and both are drawn from prior studies. This restricts generalizability.
- **Quantitative rigor**: Results are primarily qualitative (e.g., “most closely resembles”) rather than statistically benchmarked with direct effect size comparisons across models.
- **Narrow focus on fMRI signatures**: Other alignment tools (RSA, encoding models, representational dimensionality) are mentioned but not integrated into the evaluation.
- **Lack of discussion of limitations**: The absence of a dedicated limitations section weakens the transparency of the paper.
- **Overemphasis on PARROT**: Results highlight the in-house subword model as most brain-like, but the role of architecture, training data, and supervision type is not disentangled.
- **Code/data unavailability**: No resources are shared, reducing reproducibility and external validation potential.

---

### Detailed Comments

1. **Clarity and Structure**
   - The paper is well-structured
   - Language is clear and accessible, though some sections (Discussion) are brief and could expand on implications.
   - Figures are helpful, but results should be quantified more precisely (e.g., variance explained, correlation values, error bars across models).

2. **Contribution and Originality**
   - The central contribution is the use of **fMRI-inspired simulations** as a diagnostic for speech models.
   - This is original and timely, especially given ongoing debates about brain-model alignment.
   - However, the contribution is incremental unless the authors more explicitly connect these simulations to **predictive frameworks** for neuroscience (e.g., what falsifiable hypotheses about auditory coding follow from these discrepancies?).

3. **Methodology**
   - Candidate models are well-chosen, spanning supervised and self-supervised paradigms.
   - Response calculation is carefully described (unit/layer averaging).
   - NSE and normalized response metrics are reasonable, but more direct **comparative statistics** (e.g., permutation tests across model classes) would strengthen claims.
   - Suggestion: include **ablation analyses** (e.g., removing recurrent layers in PARROT, comparing different fine-tuning datasets) to better isolate drivers of alignment.

4. **Data and Analysis**
   - The study uses pre-trained models and simulations rather than new fMRI data, which is efficient but may bias comparisons toward models trained with explicit phonemic supervision.
   - Results are presented descriptively, with limited quantification of **how much variance in brain data is explained**.
   - Adding a unified comparison table (model × paradigm × brain region × alignment score) would make findings clearer.

5. **Results and Discussion**
   - Main finding (subword > word/self-supervised) is robust and consistent across both paradigms.
   - Discussion should elaborate on **why** self-supervised models (otherwise strong on behavioral tasks) underperform here. Possible explanations: task-objective mismatch, lack of explicit phonemic representations, or differences in architecture.
   - The implications for **neuroscience theory** (e.g., levels of auditory hierarchy, importance of subword-level units) should be expanded.
   - The conclusion currently reads as a summary; it should instead provide forward-looking research questions.

6. **References**
   - References are comprehensive, covering key prior work in auditory neuroscience and model-brain alignment.
   - Ensure citations to recent brain-LM alignment surveys (e.g., Tuckute et al. 2023, Schrimpf et al. 2021) are contextualized.
   - Minor inconsistencies in formatting should be fixed.

7. **Ethics and Compliance**
   - No new human data were collected; all work is based on previously published paradigms.
   - Ethical concerns are minimal. Still, a brief ethics statement (e.g., “all prior fMRI data cited adhered to ethical guidelines”) would improve transparency.

---

### Overall Recommendation
**Recommendation**: *Accept with revisions (weak accept).*

The paper presents a creative and well-motivated approach to evaluating speech recognition models through fMRI-inspired simulations. Its strengths lie in theoretical grounding, clarity, and comparative breadth. However, the work would benefit from:
- stronger **quantitative analyses** (effect sizes, statistical comparisons),
- deeper **discussion of limitations and implications**, and
- improved **reproducibility** (code/data release).

With these revisions, the paper could make a significant contribution to the literature on model-brain alignment in auditory neuroscience.

**Score:**

3

**Topic Fit:**

2